# A Poly(dA:dT) Tract in the *IGF1* Gene Is a Genetic Marker for Growth Traits in Pigs

**DOI:** 10.3390/ani12233316

**Published:** 2022-11-27

**Authors:** Weili Liao, Yifei Wang, Xiwu Qiao, Xiaoke Zhang, Haohui Deng, Caihong Zhang, Jiaqi Li, Xiaolong Yuan, Hao Zhang

**Affiliations:** 1Guangdong Laboratory of Lingnan Modern Agriculture, National Engineering Research Center for Breeding Swine Industry, Guangdong Provincial Key Lab of Agro-Animal Genomics and Molecular Breeding, College of Animal Science, South China Agricultural University, Guangzhou 510642, China; 2Guangzhou Xustoms Technology Center, Guangzhou 510623, China

**Keywords:** *IGF1*, luciferase assay, transcription regulation, poly(dA:dT) tracts

## Abstract

**Simple Summary:**

Insulin-like growth factor 1 (*IGF1*) promotes mammalian development and growth. The poly(dA:dT) tract usually acts as a promoter element to regulate gene transcription. In this study, it was found that the length of a poly(dA:dT) tract in the porcine *IGF1* promoter can regulate gene expression in vivo. Moreover, this polymorphism is associated with porcine growth traits (days to 115 kg and average daily gain). These results suggest that the poly(dA:dT) tract is a genetic marker for porcine growth traits.

**Abstract:**

Insulin-like growth factor 1 (*IGF1*) is an important regulator of body growth, development, and metabolism. The poly(dA:dT) tract affects the accessibility of transcription factor binding sites to regulate transcription. Therefore, this study assessed the effects of two poly(dA:dT) tracts on the transcriptional activity of porcine *IGF1*. The luciferase assay results demonstrated that the poly(dA:dT) tract 2 (−264/−255) was a positive regulatory element for *IGF1* gene expression, and the activities between the different lengths of the poly(dA:dT) tract 2 were significant (p<0.01). The transcription factor *C/EBPα* inhibited the transcription of *IGF1* by binding to tract 2, and the expression levels between the lengths of tract 2 after *C/EBPα* binding were also statistically different (p<0.01). Only the alleles 10T and 11T were found in the tract 2 in commercial pig breeds, while the 9T, 10T, and 11T alleles were found in Chinese native pig breeds. The allele frequencies were in Hardy–Weinberg equilibrium in all pig breeds. The genotypes of tract 2 were significantly associated with the growth traits (days to 115 kg and average daily gain) (p<0.05) in commercial pig breeds. Based on these findings, it can be concluded that the tract 2 mutation could be applied as a candidate genetic marker for growth trait selection in pig breeding programs.

## 1. Introduction

Insulin-like growth factor 1 (*IGF1*), a part of the IGF system that controls mammalian organismal growth, is a regulator of cell growth and mitogenesis in animals [1,2]. *IGF1* [3] is conserved among species; for example, porcine *IGF1* has 70 identical amino acids with those of bovine [4] and human [5]. *IGF1* regulates the growth and development of the body, mainly mediated by growth hormone (GH) [6,7,8]. A study showed that the average body weight of double *GHR/IGF1* nullizygotes is only 17% of those in normal mice [9]. The transgenic mouse offspring of *IGF1* mutation are approximately 40% smaller than the wild-type littermates [10,11]. The *IGF1* levels were positively related to growth rate and body size in dogs [12] and pigs [13]. An early study by Casas-Carrillo et al. found a QTL affecting the growth rate near the porcine *IGF1* gene [14]. Research in dog size variation demonstrated that a mutation in *IGF1* caused diversity in dog body size [15]. Elevations in mouse maternal *IGF1* abolished the normally negative relationship between fetal mass and litter size in late gestation via cross-breeding experiments [16]. Selection for post-weaning gain resulted in a greater average daily gain, and 13% greater average backfat thickness in the fast line than in the slow line [17]. The average growth hormone concentration was not significantly different, but there was a higher *IGF1* concentration in the fast line blood samples than in the slow line blood samples at about 55 kg live body weight [18]. These indicated that *IGF1* gene is a candidate gene that is associated with growth and body size in pigs. The growth traits are important because they are both breeding objectives and selection criteria in pig breeding [19].

The ubiquitous promoter element poly(dA:dT) tracts are 10–20 bp homopolymeric stretches of deoxyadenosine nucleotides (A’s), and can resist the incorporation of nucleosome assembly [20]. The existence and length of native poly(dA:dT) tracts in promotors can affect the accessibility of transcription factor binding sites near nucleosomes, thus regulating gene transcription. In yeast, poly(dA:dT) tracts strongly stimulated Gcn4-dependent activation in a length-dependent manner [21]. Various STRs with lengths of 17, 18, and 19 repeats on the background of the common haplotype C-T-T (i.e., C17TT, C18TT, and C19TT) had significantly different transcription activities for *IGF1* in Beas-2B cells [22]. The deletion of poly(dA:dT) tracts in the *AOX1* promoter could stimulate expression, while the addition of 15 bp poly(dA:dT) tracts resulted in a depression in the expression level [23]. These studies showed that poly(dA:dT) tracts with various lengths, as a member of microsatellites, might be crucial for the expression of *IGF1*.

We found two poly(dA:dT) tracts in the porcine *IGF1* promoter region. Thus, we asked whether poly(dA:dT) tracts directly regulate transcriptional activity of porcine *IGF1*, and whether it is associated with porcine growth traits. Furthermore, the study of the predicted transcription factor *C/EBPα* regulating *IGF1* expression further revealed the possible regulation mechanism of the poly(dA:dT) tract. The purpose of this study is to determine whether the polymorphism of a poly(dA:dT) tract can cause porcine growth rate variation, and whether the mutation can be used in pig breeding practices.

## 2. Materials and Methods

### 2.1. Animals, Sample Collection, and Traits Evaluated

Three Duroc and three Large White pigs were used for the collection of total DNA to clone the 5′ region of *IGF1* (Gene ID: 397491). Porcine fetal fibroblast (PFF) cells were collected as described previously [24]. The fetus was minced and digested individually in digestion media (0.25% trypsin and 0.04% EDTA) for 15 min at room temperature, followed by its dispersal in culture media containing Dulbecco’s modified Eagle’s medium (DMEM), 10% fetal bovine serum (FBS) (Gibco, Carlsbad, CA, USA), and 1% penicillin–streptomycin (Hyclone, Logan, UT, USA). The dispersed cells were centrifuged, resuspended, and cultured in culture media at 37 °C in a 5% CO_2_ atmosphere and saturated humidity.

Ear tissue samples were collected from 320 Duroc pigs, 230 Large White pigs, 22 Guanzhuang Spotted pigs, and 18 Yuedong Black pigs, raised in farms of Guangdong Province in China for polymorphism analysis. Growth traits such as birth weight, body length, average daily gain, days to 115 kg, average backfat thickness at 115 kg, and loin muscle area at 115 kg for 320 Duroc pigs and 230 Large White pigs were used for association analysis. Traits were measured as described in a previous study [25].

### 2.2. Construction of the IGF1 Promoter Luciferase Plasmid

Genomic DNA was extracted from the ear tissues of Duroc and Large White pigs using the TIANGEN Isolation/Extraction/Purification Kit (TIANGEN, Beijing, China) according to the manufacturer’s instructions. Approximately 2.7 kB of the 5′ upstream sequence of *IGF1* was PCR-amplified from pig DNA. The forward and reverse primers were 5′-ACATCCTTGCTATTTTGGTGGC-3′ and 5′-ATAACTCCCAGTGCCGAAACAA-3′. The resulting PCR product, a 2775 bp DNA sequence corresponding to the region −2467/+2 of *IGF1* (the transcription start was designated as +1), was further purified and cloned into the pMD20-T vector that was used as a template to generate a series of 5′ deletion elements using primers (Table 1). A series of 5′ deletion elements were divided into −2467/+2, −1900/+2, −1466/+2, −959/+2, −381/+2, and −100/+2. Then, they were respectively cloned into the multiple cloning site of a pGL3-basic luciferase vector between the Kpn I/Xho I sites, to be named P1, P2, P3, P4, P5, and P6. Furthermore, a series of 3′ deletion elements were divided into −381/−101, −381/−213, and −381/−284. They were constructed according to the above method, namely, P5-1, P5-2, and P5-3.

The PCR products in Large White and Guanzhuang Spotted pigs were sequenced using the P5 primer. We found a poly(dA:dT) tract with three lengths of the nucleobase T (9T, 10T, and 11T). They were constructed according to the above method, namely P5-9T, P5-10T, and P5-11T.

The P5 primer and porcine DNA (Large White and Guanzhuang Spotted pigs, 100) were used to sequence different lengths of the poly(dA:dT) tract, namely, P5-9T, P5-10T, and P5-11T. The P5-9T, P5-10T, and P5-11T vectors were constructed according to the above method.

### 2.3. Construction of the Overexpression Vector and siRNA for C/EBPα

Through the online website prediction (http://www.genomatix.de, accessed on 7 November 2020; http://www.cbrc.jp/research/db/TFSEARCH.html, accessed on 7 November 2020), it was predicted that the 5′ regulatory region of *IGF1* had *C/EBPα* transcription factor binding sites. To figure out how the poly(dA:dT) tract regulates *IGF1* transcription, the following overexpression vectors and siRNAs were constructed. *C/EBPα*-mRNA (Gene ID: 751869) from Duroc pig liver was used as a template in a PCR reaction. To clone the *C/EBPα*-mRNA, the desired sequence was amplified via PCR using a specifically designed forward primer, 5′-GGGGTACCCC AGACCAAGACTTGCCCTCCAC-3′ and reverse primer, 5′-CCGCTCGAGCGG TCTTCGGGTTTTGGTATCCTCA-3′, and ligated into the pcDNA3.1/Myc-His (-) vector. siRNA targeting *C/EBPα* was designed using siRNA-designing software (Ambion, Austin, TX, USA): siRNA#1, 5′-GCACCGGAUUGAGGAGAAA dTdT-3′, 3′-dTdT CGUGGCCUAACUCCUCUUU-5′; siRNA#2, 5′-CCAACACUGCAGAGCUCAA dTdT-3′, 3′-dTdT GGUUGUGACGUCUCGAGUU-5′; siRNA#3, 5′-GAAGAAGAGUCCUUUCAAU dTdT-3′, 3′-dTdT CUUCUUCUCAGGAAA GUUA-5′ (RiboBio, Guangzhou, China).

### 2.4. Cell Transfection and Luciferase Activity Analysis

PFF cells were maintained in the culture media, as described previously [24]. Cells were incubated at 37 °C in 5% CO_2_ to reach 80% confluence for transfection. PFF cells were cultured in 24-well plates and transfected with 0.75 μg of either P1-P6, P5-1/2/3, or P5-9T/10T/11T with pRL-TK vector containing Renilla luciferase. Furthermore, co-transfection was also carried out on PFF cells by co-transfecting P5-9T/10T/11T and *C/EBPα*. The transfection method was operated according to the instructions of lipofectamine TM LTX and PLUSTM (Invitrogen, Thermo Fisher Scientific Inc., Carlsbad, CA, USA). Luciferase activity was measured 48 h later using the Dual-Glo luciferase assay (Promega, Madison, WI, USA). The activities of different promoter fragments were expressed by detecting the ratio of firefly luciferase activity to Renilla luciferase activity [26], which allowed for the evaluation of which fragment was a *IGF1* core promoter.

### 2.5. RT-PCR and Real-Time Quantitative RT-PCR Analysis

Total RNA was purified from PFF cells using TRIzol reagent (Invitrogen, Thermo Fisher Scientific Inc., Carlsbad, CA, USA) according to the manufacturer’s instructions, and reverse transcribed using the First Strand cDNA Synthesis kit (Takara Bio Inc., Shiga, Japan). The cDNA was then diluted 1:5 in RNase-free water. Real-time PCR was performed using SYBR Green (YEASEN, Guangzhou, China). Each sample was analyzed in triplicate. Data were normalized to the expression level of *GAPDH*. The primer sequences used in PCR analysis are listed in Table 2.

### 2.6. Chromatin Immunoprecipitation (ChIP) Assays

Chromatin immunoprecipitation (ChIP) was carried out according to the instructions of the EZ-ChIP™ Chromatin immunoprecipitation kit (Millipore, Bedford, MA, USA) to reveal whether the transcription factor *C/EBPα* binds to *IGF1*. After ChIP, the DNA precipitated by the anti-*IGF1* antibody was detected with qPCR, which was conducted in a final volume of 20 μL containing 2 μL of 10 × PCR Buffer, 0.4 μL each of forward primer and reverse primers (10 μM), and 2 μL of DNA template. The primer sequences are listed in Table 2.

### 2.7. Genotyping the Simple Sequence Repeats (SSR)

Universal forward primers were labeled at the 5′ end with FAM fluorescent dyes (Shanghai Generay Biotech Co., Ltd., Shanghai, China). The amplified fragments were subjected to capillary electrophoresis in a multiload system using an ABI 3730 genetic analysis (Applied Biosystems, Darmstadt, Germany). Peaks were analyzed using GeneMarker 2.2.0 software (SoftGenetics, State College, PA, USA). GSLIZ500 was used as a size fragment standard to compared with peaks to ensure amplified fragments (Applied Biosystems). When the amplified fragments of *IGF1* gene were 378 bp, 377 bp, and 376 bp, the poly(dA:dT) tract contained 11T, 10T, and 9T, respectively. They were named 11T, 10T, and 9T, respectively. The genotype of the *IGF1* gene was also expressed using the number of T of the corresponding poly(dA:dT) tract.

### 2.8. Statistical Analysis

The Chi-squared goodness-of-fitness tests for the genotypic frequencies of *IGF1* were performed using Microsoft Excel according to Kaps and Lamberson (2009) [27].

The GLM procedure of the SAS software was used to analyze the association of different genotypes of the corresponding poly(dA:dT) tract with phenotypic variations. The trait least-squares means of different genotypes were estimated and expressed as mean ± standard error. The *p* values were adjusted with Tukey’s method, and the threshold of significant difference was p<0.05. The statistical models were as follows:(1)Y=μ+Sex+H+G+bW+e
where Y is the phenotypic value (birth weight, body length, average daily gain, day to 115 kg, average backfat thickness at 115 kg, and loin muscle area at 115 kg), *μ* is the overall population mean, *Sex* is the sex effect, *H* is the month effect, *G* is the genotypic effect, *b* is the regression coefficient, *W* is the covariate, and *e* is the random error. The *W* is live body weight when the dependent variables are loin muscle area and body length. Birth weight is used as a covariate for the analyses of daily gain and days to 115 kg. There is no covariate term when the trait of birth weight is analyzed. The random error term e is assumed to be independent and identically distributed $N\left(0,\sigma^{2}\right)$.

## 3. Results

### 3.1. Effect of Poly(dA:dT) Tracts on IGF1 Transcription Activity

There are two poly(dA:dT) tracts within the *IGF1* promoter region, namely tract 1 (−1354/−1346) and tract 2 (−264/−255) (Figure 1A). The transcription activities of P1, P2, and P3 containing tract 1 and tract 2 were significantly lower than those of P4 and P5, which only contained tract 2. The transcription activity decreased significantly when the fragment was shortened to P6 without tract 2 (−264/−255). These results demonstrated that tract 2 promoted the transcription activity of *IGF1*.

To further describe the regulations of tract 2 on *IGF1* transcription, 3′ serial deletion constructs were cloned on pGL3 vectors, and the activities of the luciferase assay were analyzed. As shown in Figure 1B, the transcription activities of P5, P5-1, and P5-2 were significantly higher than those of P5-3 in the PFF cells. These results further display that tract 2 is a positive regulatory element that plays an important role in the transcription activity of *IGF1*.

Recombinant vectors of tract 2 with various lengths were constructed and transfected into PFF cells for a luciferase assay, which could analyze the effects of tract 2 on *IGF1* transcription activity. As shown in Figure 1C, the transcription activity of tract 2, respectively containing 9T, 10T, and 11T, increased in turn. Compared with P5-9T and P5-10T, the transcription activity of tract 2 containing 11T (P5-11T) was significantly higher. The results show that the transcription activity of *IGF1* is influenced by the length of tract 2.

### 3.2. Distribution of Tract 2 Genotypes on IGF1 in Chinese Native and Commercial Breeds

We found five genotypes on tract 2 among the 553 porcine DNA samples via SSR (Figure 2). The different genotypes of tract 2 were respectively shown in Table 3 and Table 4 for native and commercial pigs. For the Chinese native breeds, three genotypes (9/9T, 9/10T, and 10/10T) of tract 2 were detected in Guanzhuang spotted pigs, and four genotypes (9/9T, 9/10T, 10/10T, and 10/11T) of tract 2 were detected in Yuedong black pigs. For commercial pigs, three genotypes (10/10T, 10/11T, and 11/11T) of tract 2 were detected. The frequency of 10/10T was highest in Duroc pigs, and the frequency of 11/11T was lowest. However, the frequency of 10/10T was lowest in Large White pigs, and the frequency of 11/11T was highest. From the allele frequencies of different pig breeds, Chinese native pigs had one more allele, 9T, compared with commercial pigs. It shows that the distribution lengths of tract 2 are different between pig breeds. The results of the $\chi^{2}$ test showed that all genetic frequency distributions were in Hardy–Weinberg equilibrium in Chinese and commercial pigs (p>0.05).

### 3.3. Association Analysis between Tract 2 on IGF1 and the Growth Traits of Commercial Pigs

To investigate the effect of tract 2 on growth traits, the association analyses between the genotypes of tract 2 and the growth traits were performed in commercial pig breeds. The results of Duroc pigs and Large White pigs are shown in Table 5 and Table 6, respectively.

The traits of birth weight, body length, average backfat thickness at 115 kg, and loin muscle area were not significantly different between the genotypes of tract 2 in Duroc pigs and Large White pigs (p>0.05). However, the effects of the genotypes of tract 2 on the traits of days to 115 kg and average daily gain in Duroc and Large White pigs were significant (p<0.05). The days to 115 kg were 5.37 d and 10.02 d shorter in the 10/10T genotype than for 11/11T genotype for Duroc and Large White pigs, respectively. Furthermore, the average daily gain was 18.69 g and 34.61 g higher in the 10/10T genotype than the 11/11T genotype for Duroc and Large White pigs, respectively.

### 3.4. Transcription Factor C/EBPα Affects the Expression Abundance of IGF1

According to the MatInspector software, a potential C/EBRα binding site within the *IGF1* promoter was located within the region −259/−245 upstream of the transcription start site. The overexpression vector of C/EBRα (pcDNA3.1-C/EBRα) was constructed and transfected into PFF cells to evaluate whether the transcription factor C/EBRα could regulate the expression of *IGF1.* After the overexpression of C/EBRα, the expression level of C/EBRα mRNA increased gradually with the enhancement of pcDNA3.1-C/EBRα concentration (p<0.01), which indicated that pcDNA3.1-C/EBRα was successfully transfected into PFF cells with a high level of expression (Figure 3A). At the same time, after the overexpression of C/EBRα, the mRNA expression level of *IGF1* in the experimental group (pcDNA3.1-C/EBRα) was significantly lower than that in the no-load control group (pcDNA3.1), indicating that *C/EBPα* could inhibit the transcription of *IGF1* (Figure 3B).

To assess the effects of the length of tract 2 on the transcriptional activity of *IGF1* after the overexpression of C/EBRα, we co-transfected cells with pcDNA3.1-*C/EBPα* and P5-9T, P5-10T, or P5-11T, respectively (Figure 3C). The activity of co-transfection with P5-11T was the highest, followed by P5-10T and P5-9T (p<0.01). Furthermore, the activities of the co-transfected cells were 5- to 6-fold lower than those without *C/EBPα* (Figure 1C). The result also showed that *C/EBPα* inhibited the transcription of *IGF1*. The lengths of tract 2 affect the transcription activity of the *IGF1* gene regulated by *C/EBPα*.

To further reveal the transcriptional regulation of *C/EBPα* on *IGF1*, a ChIP assay was used to verify whether the transcription factor *C/EBPα* specifically binds to the cis-acting element (tract 2) in the *IGF1* promoter. As shown in Figure 3D, the cis-acting element (tract 2) in *IGF1* was indeed bound with *C/EBPα*. The results indicated that *C/EBPα* could bind to tract 2 to participate in the regulation of *IGF1* gene expression, thus inhibiting the expression of *IGF1*.

## 4. Discussion

It is known that the poly(dA:dT) tracts within promoters can regulate transcription [21], and that the effect sizes are affected by the length of the poly(dA:dT) tracts and the distance between the poly(dA:dT) tracts and transcription factor sites [23,28]. Our results in Figure 1A,B showed that the poly(dA:dT) tract 2 in the porcine *IGF1* gene promoter located within the core promoter region affects *IGF1* gene expression. The length of tract 2 would change the nucleosome organization [20], thus influencing the accessibility of the transcription factor. *C/EBPα* belongs to the C/EBP family with growth regulatory activity. Various C/EBPs are specific to the promoter regulation element of the *IGF1* gene. The C allele of rs35767 in the human *IGF1* gene provides a binding site for C/EBPD, which is essential for the gradational transactivation property of eSTR to activate *IGF1* promoter activity [29]. Fermented feed significantly enhances the binding of the C/EBPβ and *IGF1* promoter and promotes the expression and production of *IGF1* in liver, thus promoting the growth of pigs [30]. The *C/EBPα* binding site was predicted in the region of −259/−254 in the porcine *IGF1* promoter. It was found that transcription factor *C/EBPα* participates in the regulation of *IGF1* expression by binding to tract 2 (Figure 3), thus inhibiting the transcriptional activity of *IGF1* gene in vitro. Our study shows that *C/EBPα* is a transcription factor of the *IGF1* gene, and the length of tract 2 has a significant impact on the binding of transcription factor *C/EBPα*.

On the grounds that the luciferase activities of DNA fragments containing the tract 2 of the *IGF1* gene differed highly significantly (Figure 1C and Figure 3), the lengths of tract 2 in the *IGF1* gene promoter can be a causal mutation. The mutation can change the *IGF1* expression. Because of the *IGF1* levels related to growth rate and body size in animals [12,13], the tract 2 in the *IGF1* gene promoter might associate with the growth traits of pigs.

The allele 9T is unique for Chinese pigs, and the alleles of 10T and 11T are common to all breeds (Table 3 and Table 4). Some genotypes were not detected because of their small sample sizes in Yuedong Black pigs. It is well known the two Chinese indigenous pigs have lower growth rates and smaller body sizes than those of commercial breeds [31]. The Chinese breeds are conservation populations and do not experience modern artificial selections, while the foreign breeds are selected for faster growth rates. However, the allele frequencies of the poly(dA:dT) tract in all breeds are in Hardy–Weinberg equilibrium, illustrating that this genetic polymorphism is not seriously affected by artificial selection. The allele equilibrium also guarantees the association analysis results are unbiased. Hence, this poly(dA:dT) tract can be a major gene, but it cannot be a determining factor for porcine growth rates. Because domestic pigs do not have performance records, the association analyses were only conducted with Duroc and Large White pigs. In Table 5 and Table 6, the genotype 10/10T pigs had the highest average daily gains and the lowest days to 115 kg live weight. It had been shown that the *IGF1* expression level is positively related with growth rate [13,14]. That the average daily gains between the three genotypic in pigs differed significantly (p<0.05) demonstrated that the mutation of tract 2 contributes to *IGF1* expression in vivo. However, the transcriptional activity of *IGF1* with genotype 11/11T was highest in vitro, which is different with the result in vivo. These results reflect the complexity of *IGF1* expression regulation and need further study on this issue.

A previous study has showed that the polymorphism of CA repeats microsatellites near the tract 2 in the *IGF1* promoter was significantly associated with plasma *IGF1* concentration in pigs. The longer genotype of CA displayed a higher live weight in Landrace boars, a higher carcass weight in Duroc [32], and a higher average daily gain in Large White [33], but there was no clear relationship between the CA repeats and growth rate in Shanxi White pigs, a Chinese domestic breed [34], and these observations suggested that the CA repeats in the *IGF1* promoter are important elements that regulate the transcription and function of *IGF1* in pigs. Moreover, the CA repeats are near the tract 2 in the *IGF1* promoter, indicating that CA repeats may link with the poly(dA:dT) tract in pigs, and it was likely that the effects of the CA microsatellite on porcine performance are due to the linkage with the poly(dA:dT) tract. Therefore, further works are needed to explore the correlations between the CA repeat microsatellite and the poly(dA:dT) tract in pigs.

The proportions of the tract 2 genotypes in Duroc are significantly different from those in Large White (Table 4). Previous studies have suggested that the maternal *IGF1* stimulates prenatal growth and the development of the conceptus [35,36]. It is inferred that *IGF1* may be associated with porcine reproduction performance. Duroc is a paternal line and is selected for growth and carcass traits, and Large White is a maternal line and is selected for reproduction traits. Hence, the discrepancy of genotypic distributions might be results and responses to different selection objectives in pigs.

In summary, the poly(dA:dT) tract 2 in the *IGF1* gene promoter affects the growth rates of pigs. These results will advance our understanding of the genetic basis of the growth traits in pigs. In addition, the identified poly(dA:dT) tract will be useful for the genetic improvement of daily gains in pig breeding.

## Figures and Tables

**Figure 1 animals-12-03316-f001:**
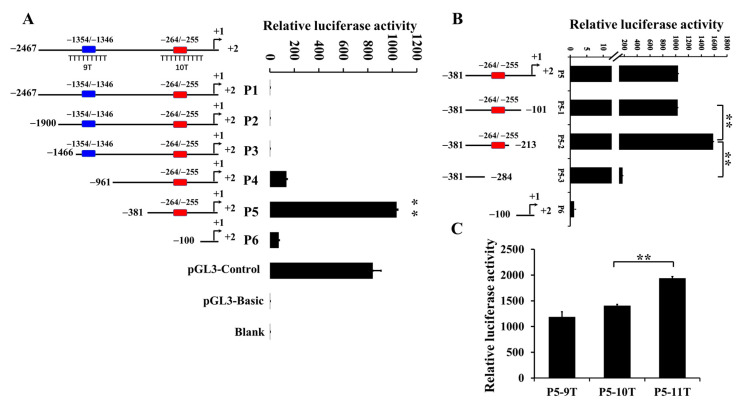
Analysis of poly(dA:dT) tracts on *IGF1* promoter activity in pigs. Luciferase report vectors containing various lengths of the *IGF1* promoter region were constructed and transfected into porcine fetal fibroblast cells. (**A**): Analysis of poly(dA:dT) tracts on *IGF1* transcription activities detected using luciferase assay. Tract 1: (−1354/−1346); tract 2: (−264/−255); blank: untreated cells; pGL3-Basic: negative control; pGL3-Control: positive control; (**B**): Effect of tract 2 on *IGF1* transcription activity. (**C**): Detection of the transcription activities of different genotypes of tract 2 on *IGF1*. Data were presented as mean ± sd. The data shown were three independent experiments. A two tailed t-test was used to determine the statistical significance of the difference between the promoter constructs, and ** was annotated as *p* < 0.01.

**Figure 2 animals-12-03316-f002:**
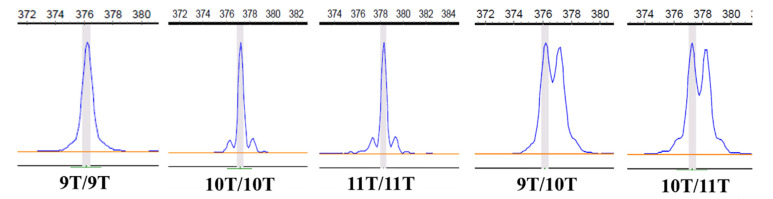
Fluorescence capillary electrophoresis map of genotypes for tract 2 on *IGF1* polymorphisms.

**Figure 3 animals-12-03316-f003:**
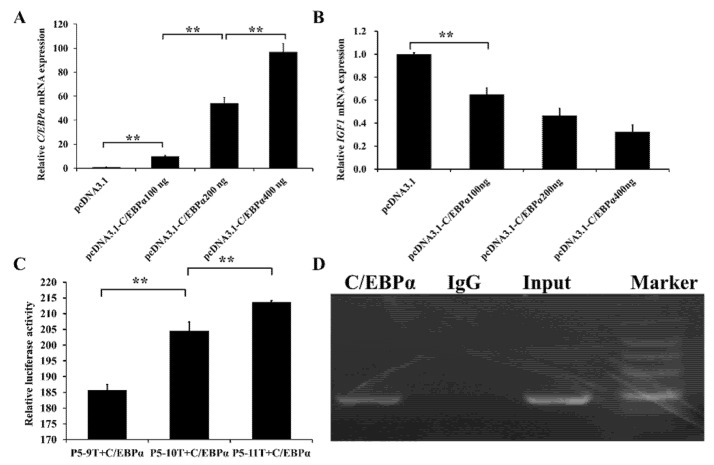
*C/EBPα* is critical for the expression of *IGF1.* (**A**): mRNA expression of *C/EBPα* at different concentration gradients of pcDNA3.1- *C/EBPα*. (**B**): mRNA expression of *IGF1* at different concentration gradients of pcDNA3.1- *C/EBPα*. (**C**): Luciferase assay of PFF cells co-transfected with different tract 2 genotypes and *C/EBPα*. (**D**): Binding of *C/EBPα* sites to *IGF1* in vitro detected using ChIP. The input lanes correspond to PCR products derived from chromatin prior to immunoprecipitation. The IgG lanes correspond to PCR products containing chromatin immunoprecipitated with antibodies against control IgG. The *C/EBPα* lanes correspond to PCR products containing chromatin immunoprecipitated with antibodies against *C/EBPα*. Marker indicates DNA 2000 marker. ** was annotated as *p* < 0.01.

**Table 1 animals-12-03316-t001:** Primers used for *IGF1* promoter reporter construction.

Construct	Primer	Sequence (5′-3′)
P1 (−2467/+2)	Sense-2467	GGGGTACCCC CTGTTGCTGGCTCGCTCTACCC
P2 (−1900/+2)	Sense-1900	GGGGTACCCC AGATGGGTGCAGTTCTTCAGCT
P3 (−1466/+2)	Sense-1466	GGGGTACCCC CACCACATGACAGTGACGTTTT
P4 (−959/+2)	Sense-959	GGGGTACCCC ATCTCCTACTTCGCAAAACCAA
P5 (−381/+2)	Sense-381	GGGGTACCCC CCCAGCACTGTCTTCCAATCTA
P6 (−98/+2)	Sense-98	GGGGTACCCC AAAATGCTTCTGTGCTCTAGTT
	Antisense	CCGCTCGAGCGG CCCTCTTCTGGCAAAGTTATCG

**Table 2 animals-12-03316-t002:** Primers used for PCR.

Gene	Sequence (5′-3′)	Product
*IGF1*	F: TGCGGAGACAGGGGCTTTT	154 bp
	R: ACTTGGCAGGCTTGAGGGGT	
*C/EBPα*	F: ATGAGCAGCCACCTCCAGAGCC	168 bp
	R: CGGGTCGATGTAGGCGCTGATGT	
*IGF1*-ChIP	F: CCTGCGCAATGGAATAAAGT	163 bp
	R: ATTGGGTTGGAAGACTGCTG	

**Table 3 animals-12-03316-t003:** Genotypic frequency of tract 2 on *IGF1* gene among Chinese native pigs.

Breed	N	Genotype Frequency				Observation Value		Theoretical Value		$\chi^{2}$	*p*
		9/9T	9/10T	10/10T	10/11T	9/10T	Others	9/10T	Others		
Guanzhuang Spotted pigs	22	0.18	0.55	0.27	0	12	10	10.91	11.09	0.26	0.61
Yuedong Black pigs	18	0.06	0.56	0.33	0.06	10	8	7.67	10.33	1.64	0.27

Note: N: number of genotyped pigs. Others: 9/9T, 10/10T, and 10/11T.

**Table 4 animals-12-03316-t004:** Genotypic frequency of tract 2 on *IGF1* among commercial pigs.

Breed	N	Genotype Frequency			Observation Value			Theoretical Value			$\chi^{2}$	*p*
		10/10T	10/11T	11/11T	10/10T	10/11T	11/11T	10/10T	10/11T	11/11T		
Duroc	328	0.48	0.39	0.13	157	128	43	148.91	144.19	34.91	4.13	0.13
Large White	225	0.02	0.33	0.65	5	74	146	7.84	68.32	148.84	1.56	0.46

Note: N: number of genotyped pigs.

**Table 5 animals-12-03316-t005:** Least-squares analysis of *IGF1* genotypes and growth traits in Duroc Pigs.

Traits	Genotype (N)		
	10/10T (156)	10/11T (129)	11/11T (43)
Birth weight, kg	1.67 ± 0.08	1.71 ± 0.02	1.69 ± 0.04
Body length, cm	117.25 ± 0.31	117.01 ± 0.34	116.86 ± 0.59
Average daily gain, g/day	636.55 ± 2.87 ^a^	629.30 ± 3.16 ^ab^	617.86 ± 5.54 ^b^
Days to 115 kg, day	179.20 ± 2.63 ^a^	180.38 ± 2.54 ^ab^	184.57 ± 2.99 ^b^
Average backfat thickness at 115 kg, cm	9.69 ± 0.12	9.77 ± 0.13	9.54 ± 0.22
Loin muscle area, cm^2^	42.63 ± 0.38	42.59 ± 0.42	41.92 ± 0.74

^a,b^ Represent statistically significant differences at a level of p<0.05.

**Table 6 animals-12-03316-t006:** Least-squares analysis of *IGF1* genotypes and growth traits in Large White Pigs.

Traits	Genotype (N)		
	10/10T (5)	10/11T (75)	11/11T (145)
Birth weight, kg	1.48 ± 0.12	1.78 ± 0.04	1.65 ± 0.03
Body length, cm	123.54 ± 1.57	120.48 ± 0.43	120.49 ± 0.31
Average daily gain, g/day	721.15 ± 16.25 ^a^	679.47 ± 4.43 ^b^	686.54 ± 3.21 ^ab^
Days to 115 kg, day	157.31 ± 4.04 ^a^	167.33 ± 1.10 ^b^	165.52 ± 0.80 ^ab^
Average backfat thickness at 115 kg, cm	14.98 ± 1.06	15.17 ± 0.29	14.79 ± 0.21
Loin muscle area, cm^2^	37.68 ± 1.73	40.77 ± 0.47	40.58 ± 0.34

^a,b^ Represent statistically significant differences at a level of p<0.05.

## Data Availability

The performance data are not available due to farmers’ disagreements.

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
