# Peer review of "A Poly(dA:dT) Tract in the IGF1 Gene Is a Genetic Marker for Growth Traits in Pigs"

_animals, 2022, doi:10.3390/ani12233316_

Round 1
Reviewer 1 Report
Lines 51-55 : looks like result and may be replaced with a clear cut objective and potential benefit of this study
Lines 65-67: Studies has been made with unequal number of observation and 22 Guanzhuang spotted pigs and 18 Yuedong black pigs were studied and may not give the required statistical conclusion
lines 105-115 methods need reference citation
Lines 143 to 144 from According ..........pigs may be deleted
Line 144 : "SAS software" may be started with "The SAS software....."
Lines 147-148 : and the least square mean value of the traits was esti-147 mated and expressed as mean ± standard error. The standard of significant difference 148 was ?<0.05. may be modified as : "and the least- squares analysis was made with GLM program of SAS (reference citation) and means were compared using ..... mention the post hoc test used
lines 150-155: is the same model used for all the traits. The birthweight has been analysed as a parameter and how can you use this model for analysing the birthweight. No information on when the body length measured and the reason for using birthweight as a covariate for body length may be indicated. The model used for association studies need revision and revised analysis.
Line 162 : "These results demonstrate ............" may be modified as "These results demonstrated that ...."
Table 3 may be removed as the information made with few samples
Table 5 and Table 6 used different decimals for the birth weight and a uniformity may be made. Only overall mean alone given and kindly give the individual effect means also
The "least square" in different places may be modified as Least-Squares
line 236: To assess the effects of lengths of tract 2 may be modified as "To assess the effect of length of tract 2....."
line 241: result also shows that may be modified as "result also showed that........"
line 246: "the results indicate" may be modified as "results indicated"
In general the result may be given in past tense
Discussion is not comprehensive and need major revision. the lines 290 to 294 are just results and may be supplemented with suitable discussion with cited references.
The association analysis and discussion not properly made and need revised analysis and more detailed discussion with cited references
The summary may be revised based on the revised results
Author Response
Thank the experts for your valuable comments on our paper. We have completed all revisions to the review comments. Please refer to word for detailed revisions.

Reviewer 2 Report
The manuscript fits within the scope of Animals.
The authors do a nice jobs discussing relevant work in the discussion but it is necessary to include in the discussion significant reference: Analysis of Association Between a Microsatellite at Intron 1 of the Insulin-Like Growth Factor 1 (IGF1) Gene and Fat Deposition, Meat Production and Quality Traits in Italian Large White and Italian Duroc Pigs (Fontanesi et al. 2013, Italian Journal of Animal Science, Volume 12, Issue 3.). Properties of fat deposition respectively back fat thickness Large White and Duroc Pigs.
Author Response

(The authors gave the same response as above.)

Reviewer 3 Report
Reports of the review: A Poly (dA:dT) Tract in IGF1 Gene is a Genetic Marker for Growth Traits in Pigs
The paper entitled “A Poly (dA:dT) Tract in IGF1 Gene is a Genetic Marker for Growth Traits in Pigs” presents the effect of a poly (dA:dT) tract in the IGF1 gene expression and in growth traits in pigs. I found the topic interesting specially because of the detailed description of the poly (dA:dT) and the results found.
As strengths of the manuscript, I highlight the novelty, the methodology used, and the presentation of the results. As limitations I can highlight the scale of the introduction, that could explore more the topic specially for clarifying the use of the poly (dA:dT) and more details in the material and methods. In this sense, I recommend some minor revisions on the manuscript to clarify some points to the readers of the manuscript.
Some specific suggestions are below:
Title
L2: Please include “the” before “IGF1” in the title of the manuscript.
Simple Summary
L13: Please change “it is found” to “it was found”.
L14: Shouldn’t “in vivo” be in italic?
L14: Please change “the polymorphism” to “this polymorphism”.
Abstract
L17: Please replace semicolon by comma.
L18: Please include oxford comma after “development”.
L18: Please remove “usually”.
L19: Consider include “Therefore, this study assessed […]” ate the beginning of the sentence.
L21-22: Please replace “[…]; and the activity between lengths of poly (dA:dT) tract 2 differed highly significantly (?<0.01).” by “[…]; and the activity between different lengths of the poly (dA:dT) tract 2 was significant (?<0.01).”.
L23: Would be important include what C/EBPα is.
L23: Please include “the” after “inhibited”
L23-24: Please replace “[…]; and expression levels also differed 23 highly significantly between lengths of tract 2 after C/EBPα binding.” by “[…]; and the expression levels between lengths of tract 2 after C/EBPα binding were also statistically different (P<?).”.
L24-25: Consider changing “Only 10T and 11T alleles in 24 tract 2 were found in commercial pig breeds […]” to “Only the alleles 10T and 11T were found in the tract 2 in commercial pig breeds […]”.
L25: Please replace the semicolon by comma and include the oxford comma after “10T”.
L27: Please replace “was” by “were”.
L28: Please replace the semicolon by comma.
L29: Please be clear in which mutation you are talking about to match the objective; consider include the name of the mutation.
Introduction
L37: Please include “A” before “Study”.
L40-41: I recommend to remove the significance levels of the citations.
L43-45: Which specie you are talking about in those cross-breeding experiments?
L45-46: I don’t’ think you have provided enough information in this paragraph to say that IGF1 is a “candidate gene” for growth and body size in pigs. Only one reference was in pigs and nothing about how conserved across species the IGF1 gene is was mentioned. I recommend include more refences in pigs and how conserved IGF1 is across mammals. As you are also talking about commercial pigs, including citations about how important the IGF1 would be for selection is also suggested.
L47-50: I found this paragraph too much summarized. Please include more information about the poly (dA:dT) tracts. What are this poly (dA:dT) tracts? How many of them are described in the literature? Why you have explored them in this research? Are there any previous reports of of similar study/use of poly (dA:dT) tracts in pigs or other species?
L52: Please replace “were” by “was”.
L53-54: Please replace “[…] and association analyses were done to see whether the poly (dA:dT) tracts affect porcine growth traits.” by “[…] then association analyses were carried out to evaluate the effect of the poly (dA:dT) tracts in porcine growth traits.”.
L51-55: Please highlight the importance of growth traits in pigs and how this study can could contribute to the literature in the subject.
Materials and Methods
L57: Please change the subsection name to “Animals, Sample Collection, and traits evaluated”
L58: Please inform how many Duroc and Large White pigs were used for collection of total DNA.
L63: Please include brand, city, and country for “FBS” and “penicillin-streptomycin”.
L66: “White” instead of “white”.
L67: Please include the oxford comma after “spotted pigs”.
L69: Wouldn’t be “days to 115 kg”?
L69: Please double check if “average backfat thickness to 115 kg” is correct.
L71: Please include “in a previous study” after “described”.
L75-76: Please rephrase the sentence “Approximately […] approach.”.
L79: IGF1 must be in italic.
L84-85: Please inform what “P5-1, P5-2, and P5-3” mean. Note I included “and” before “P5-3”, do the same in the manuscript.
L86: Please inform what “P5-9T, P5-10T, and P5-11T” mean, specifically, the end “9T, 10T, and 11T”.
L91: Please include in this section the purpose of constructing and overexpression vector and siRNA for C/EBPα.
L100: Please include brand, city, and country for “siR-99 NA-designing software”.
L106: Please include the reference for the first sentence of this paragraph.
L111-112: Please include brand, city, and country for “lipofectamine TM LTX and PLUSTM”.
L113: Please include brand, city, and country for “Dual-Glo luciferase assay”.
L115: Please replace “[…] which could analyze which fragment was IGF1 core promoter.” by “[…] which allowed evaluating which fragment was a IGF1 core promoter.”
L117: Please include brand, city, and country for “TRIZOL Reagent”.
L118-119: Please include brand, city, and country for “1st Strand cDNA 118 Synthesis kit”.
L119: Please rephrase the sentence “The cDNA […] water”.
L118: Please include brand, city, and country for “SYBR green”.
L124: Please include in this section the purpose of ChIP analysis.
L126: Please include brand, city, and country for “EZ-ChIP™ Chromatin immunoprecipitation kit”.
L127: IGF1 must be in italic.
L132-133: Please include brand, city, and country for “FAM fluorescent dyes”.
L134: Please include brand, city, and country for “ABI 3730 genetic analysis”.
L138: Please include the oxford comma after “377 bp”.
L143-146: Please rephrase the sentence “According […] gene.”.
L146-148: Please split the sentence “The genotype […] error.” in two sentences.
L148-149: What was the significance test used? F? t-student? Tukey?
L151: Replace “program” by “procedure”.
L151: Inform if this is a statistical or matrix model. Also, if it is a statistical model please include the indexes of the effects appropriately, if it is a matrix model use the appropriate notation for matrices and vectors.
L153: Please inform how the genotypic effect was treated (fixed or random). I recommend do the same for the other effects and provide the assumptions for the distribution of the random effects.
Results
L166: Please include the oxford comma after “P5-1”.
L203 and L205: Please include the p-value for the assigned chi-square values in the tables 3 and 4.
L228: Please replace “see” by “evaluate”.
Discussion
L263: Please replace “[…] is an effect factor to IGF1 gene expression.” by “[…] affects the IGF1 gene expression.”
L284: It would also be important talking about the sample size used to estimate the allelic frequencies specially for the Chinese breeds when presenting the results of Hardy-Weinberg equilibrium.
L286-288: I believe you need more information to say that it is a major gene. At least estimate the variance explained by the region and compare with other regions.
L295-296: It would be important including a paragraph to talk about the limitations of the work done and future steps. It is also important to highlight how the industry can benefit from this study as growth traits are very important and commercial breeds were used.
Author Response

(The authors gave the same response as above.)

Round 2
Reviewer 1 Report
Dear Sir
The authors corrected as per the suggestions and my be be published after making refinement of the lines between 357-372 for better meaningful communication to the audience
Author Response
Thank the expert for your valuable comment on our paper. We have completed all revisions to the review comments. Please refer to word for detailed revisions.
